# Disciplinary Approaches for Cannabis Use Policy Violations in Canadian Secondary Schools

**DOI:** 10.3390/ijerph18052472

**Published:** 2021-03-03

**Authors:** Megan J. Magier, Scott T. Leatherdale, Terrance J. Wade, Karen A. Patte

**Affiliations:** 1Faculty of Applied Health Sciences, Brock University, 1812 Sir Isaac Brock Way, St. Catharines, ON L2S 3A1, Canada; twade@brocku.ca (T.J.W.); kpatte@brocku.ca (K.A.P.); 2School of Public Health and Health Systems, University of Waterloo, Waterloo, ON N2L 3G1, Canada; sleatherdale@uwaterloo.ca

**Keywords:** cannabis, policy, discipline, schools, youth

## Abstract

The objective of this study was to examine the disciplinary approaches being used in secondary schools for student violations of school cannabis policies. Survey data from 134 Canadian secondary schools participating in the Cannabis use, Obesity, Mental health, Physical activity, Alcohol use, Smoking, and Sedentary behaviour (COMPASS) study were used from the school year immediately following cannabis legalization in Canada (2018/19). Despite all schools reporting always/sometimes using a progressive discipline approach, punitive consequences (suspension, alert police) remain prevalent as first-offence options, with fewer schools indicating supportive responses (counselling, cessation/educational programs). Schools were classified into disciplinary approach styles, with most schools using Authoritarian and Authoritative approaches, followed by Neglectful and Permissive/Supportive styles. Further support for schools boards in implementing progressive discipline and supportive approaches may be of benefit.

## 1. Introduction

Little is known about what the approaches secondary schools are using to prevent student cannabis use, especially post legalization in Canada on 17 October 2018. At a minimum, school polices regarding cannabis prohibit use on and near school grounds [1]; however, limited research has examined how schools respond when students violate these policies. Identifying what disciplinary styles are being used across Canada is a necessary first step in determining how to effectively minimize youth-onset cannabis use and to guide evidence-based decision making in secondary schools moving forward. In order to provide a current picture of the disciplinary environment in secondary schools, this study aimed to examine what approaches secondary schools across different provinces in Canada reported using when students violate school cannabis policies, in the school year immediately post legalization.

Bill C-45 set a nationally regulated provision on cannabis production, distribution, retail, possession and consumption, however, provinces and territories were still tasked with developing additional legislation and policies [2]. Even though cannabis use is only legal among adults (18, 19, or 21 years and over, depending on the province or territory), youth may be indirectly affected through de-stigmatization of use and potential changes in accessibility [3,4]. Youth in Canada use cannabis at the highest rates globally [5]. After a steady decline in youth cannabis use, a gradual increase has since occurred from the beginning of federal discourse around legalization, particularly in intermittent use [6].

Traditionally, it was common for schools to use a “zero tolerance” approach to student misconduct, which aims to send a deterrent message to other students and immediately remove high-risk students. Some evidence suggests expulsion policies can have unintended negative effects on students [7,8]. Punitive approaches have traditionally been used to scare students into compliance, but research suggests these tactics can further alienate students that need help, potentially increasing their likelihood of substance use, drop out, delinquent behaviour, and leading to poorer mental health and wellbeing outcomes [9,10,11,12]. Instead, there has been a move toward progressive disciplinary approaches, in which sanctions get progressively stronger with subsequent offences, with the goal of promoting favourable decision making and offering a more supportive environment [13].

Stemming from Baumrind’s *Theory of Parenting*, three clusters of disciplinary styles have been described: authoritative, authoritarian, and permissive [14]. Baumrind found that there were two types of permissive parents, those who do not wish to inhibit their children, and parents who avoid responsibility of their children. Authoritative parenting was characterized by a high level of demandingness (i.e., high standards) and responsiveness (i.e., open communication) for their child, whereas authoritarian parents had a high demandingness but low responsiveness [14]. Applying disciplinary styles to schools, authoritarian schools are described as taking a “zero-tolerance”, highly structured, and controlling disciplinary approach, without attempts to understand the circumstances that contributed to misconduct [15]. On the other hand, authoritative schools offer balance between enforcement of the rules and responsiveness to students’ needs [15].

To classify the disciplinary environments in schools, Cornell and Huang (2016) designed an Authoritative School Climate score (ASC) based on student perceptions of the disciplinary structure and support offered in each school. Disciplinary structure was defined by whether students perceived school rules as fair and reasonable [1]. A higher disciplinary structure meant that students were able to explain themselves and were punished fairly. Many benefits have been associated with a higher authoritative climate score such as higher educational aspirations [16,17], fewer problem behaviours [18], and higher grades, relative to schools with a lower score [17,19]. In support, the International Youth Development Study in Washington State, USA and Victoria, Australia surveyed students in grades 7 and 9 and found that the use of out-of-school suspensions predicted increased cannabis use, while teacher counselling resulted in a 50% reduction in cannabis use rates; however, reporting students to a nurse or counsellor had no effect on later cannabis use [3].

In 2009, the Ontario Ministry of Education mandated all boards to require their schools to develop and implement a progressive disciplinary policy [20,21]. Principals are advised to consider a *range of options*—including prevention programs, interventions, consequences (e.g., an assignment, detention), and supports (e.g., a conversation with the student, counselling from a social worker)—to determine the most appropriate response to each situation, taking into consideration various mitigating factors (e.g., students’ history and stage of growth and development, the nature and severity of the behaviour, the impact of the behaviour on the school climate) [20]. In addition, school boards are suggested to actively engage parents and community organizations (e.g., social services and mental health agencies) in ongoing conversations as partners. In line with authoritative approaches, under this policy, schools are to provide students with opportunities to reflect on and learn from their own actions to prevent reoccurrence, with more serious consequences (e.g., suspension or expulsion) reserved for when inappropriate behaviours escalate or are repeated. However, the Ontario Education Act stipulates that in serious student incidents, more punitive options along the continuum of progressive discipline (e.g., suspension) may be required, with the possession of cannabis, being under the influence of cannabis (“unless the pupil is a medical cannabis user”), and giving cannabis to a minor, on school property included as examples of serious incidents [22].

Despite government mandated policies and procedures in certain regions, there remains limited evidence on what approaches are being used by schools. Most schools have consequence measures for the use of cannabis on school property or during school hours, although differences in school-to-school disciplinary approaches exist. In the school year immediately prior to cannabis legalization in Canada, the vast majority of secondary schools reported confiscating the product, informing parents, alerting police, and suspending students from school, among their first offence disciplinary response options, while few schools indicated less punitive options of requiring students to help around the school or assigning additional class work [23]. Despite many schools indicating punitive first offence consequences, most schools reported always using the progressive discipline approach, which was associated with lower reports of student cannabis use [15]. School responses to cannabis may have shifted since legalization of use among adults, potentially becoming less punitive and more supportive with changes in social norms, or the reverse, given increased attention to the prevention of use among adolescents. To the best of our knowledge, no study has classified school disciplinary approach styles based on the polices or procedures assessed at the school level. Therefore, this study is the first to establish a framework for classifying schools into disciplinary approach styles based on first-offence response measures reported by schools in the year immediately following cannabis legalization.

## 2. Materials and Methods

### 2.1. Design

This study used data from Year 7 (2018/19 school year) of the Cannabis use, Obesity, Mental health, Physical activity, Alcohol use, Smoking, and Sedentary behaviour (COMPASS) Study. COMPASS is an ongoing (started in 2012/13) prospective study designed to collect hierarchical data from a rolling cohort of students in grades 9 through 12 and the secondary schools they attend [24]. Schools were purposefully selected based on permitted use of active-information passive-permissoin parental consent protocols for the collection of student-level data. Information letters and recruitment packages were sent to school boards, and then to individual schools, after board approval and according to the requirements outlined by their board [25]. COMPASS has received approval from the University of Waterloo Human Ethics Committee, the Brock University Research Ethics Board, and all participating school boards. A full description of the COMPASS design is available in print [24] or online (www.compass.uwaterloo.ca, accessed on 2 March 2021).

### 2.2. Data Collection

COMPASS uses an online School Policy and Program (SPP) scan tool to gather information on the programs, policies, and protocols present within the school related to student health, as well as if changes to the programs and policies have been made over time. The SPP is completed once annually at the same time as the school’s student data collection. A member of the school administration that is most familiar with the programs, policies, and protocols in the school is identified and sent an email with a link to the online survey. Schools are also encouraged to consult other staff members and have a small group complete the SPP. A paper copy is provided if preferred. Schools are also asked for a copy of their school policy handbook. The SPP was based on a previously validated Healthy School Planner tool [24,25,26,27] and has an annual response rate of 100% of participating schools. If any missing, incomplete, or ambiguous responses on the SPP are identified after data collection, study staff follow up with school contacts by phone to clarify.

### 2.3. Sample

In Year 7 (2018/19), 136 secondary schools in British Columbia (BC) (n = 15), Alberta (n = 8), Ontario (n = 61), and Quebec (n = 52) participated, including both private and publicly funded schools, across rural to large urban areas.

### 2.4. Measures

The *first-offence consequences for school cannabis use policy violations* were assessed by the question: “What are the consequences for a first offence for students who are caught violating your school’s written policies or practices on marijuana? (Check all that apply)”. Schools were categorized based on the potential first-offence disciplinary consequences from the “check all that apply” question. Categories were based on literature from restorative, restitutive, and punitive disciplinary approaches [7,28], the Ontario Progressive Disciplinary Policy [20], which advises a range of consequences and supports, and research assessing school climate scores by applying Baumrind’s *Theory of Parenting* [14]. Previous studies have scored school climate based on student responses to measures of demandingness and responsiveness about their school [1,17,28,29,30]. Response options to first-offence violations of school cannabis policies were categorized as follows:Punitive Consequences (3 items; scored 0–3): “Alert police”; “issue a fine”; “out-of-school suspension”.Supports (i.e., restorative) (3 items; scored 0–3): “Encourage but not require an assistance, education, or cessation program”; “require to participate in an assistance, education, or cessation program”; “refer to a counsellor”.Mild Approaches (2 items; scored 0–2): “Assign additional class work”; “assign work around school”.Moderate Approaches (2 items; scored 0–2): “Detention” and “in-school suspension”.Other (4 items; scored 0–4): “Give warning”; “refer to administrator”, “confiscate substance”, and “inform parents”.

Each school was scored for the number of first-offence response options indicated in each of the five categories. Two Ontario schools did not respond to the question regarding first-offence disciplinary approach violations to cannabis use and where removed, resulting in a total of 134 schools. Due to the lack of clear categorization for all first-offence disciplinary response options, the category “other” was developed to encompass options that do not simplistically fit within one of the other categories. Based on the scores that schools received for each category, each school was classified into one specific *disciplinary approach style* as follows:“Authoritarian” was defined as scoring high in punitive first-offence disciplinary approaches (≥2 items), low in supportive approaches (<2 items), and reporting any number of moderate, mild, or other approaches.“Authoritative” was defined as scoring high in both punitive (≥2 items) and supportive approaches (≥2 items), low in moderate (<2 items) and mild approaches (<2 items), and any number of other approaches.“Neglectful” was defined as scoring low in punitive (<2 items) and supportive (<2 items) approaches, low mild approaches (<2 items) and any number of moderate approaches and other approaches.“Permissive/Supportive” was defined as scoring low in punitive (<2 items), moderate (<2 items), and mild approaches (≤2 items), high in supportive approaches (≥2 items), and any number of other approaches.“Other” disciplinary approach style was defined as having a mixed approach, with two supportive approaches, one punitive approach, two moderate approaches, three other approaches, and no mild approaches.

Schools were also classified as to whether they use a *progressive disciplinary approach* for substance use policy violations, by asking: “Do sanctions get stronger with subsequent violations of alcohol and marijuana use (i.e., progressive discipline approach)?”, with the provided response options “always”, “sometimes”, and “never”.

### 2.5. Statistical Analysis

All analysis was conducted using SAS 9.4. First, descriptive statistics were calculated for school administrator responses to the school disciplinary measure, which used a “check all that apply” response style for a list of 14 disciplinary options for student first-offence violations of school cannabis policies. Specifically, descriptive statistics were used to explore the frequency, mean, distribution, and range that each of the 14 discipline response options were indicated by schools. Differences were explored by school province. Second, each school was scored according to the number of first-offence cannabis policy violation response options they indicated in each of the categories described above (i.e., punitive, moderate, mild, supports, and other). Third, the number of first-offence response options in each category was used to classify each school into a school disciplinary approach style (i.e., Authoritarian, Authoritative, Neglectful, Permissive/Supportive, and Other), according to the above criteria. Lastly, schools were classified according to whether they indicated using a progressive disciplinary approach for subsequent violations.

## 3. Results

Out of 134 schools, the majority were classified as having an enrolment within 0–500 students (47.8%). The most common school area median household income was CDN 25,000–75,000 (74.6%) and most schools fell within a medium/large urbanicity range (93.0%).

Figure 1 shows the number of schools indicating using various first-offence response options when students violate school cannabis use policies. The mean number of first-offence disciplinary approaches reported by schools was 7.06 (SD = 2.04), when asked to “check all that apply” in a list of 14 possible options. All schools selected using at least one type of first-offence approach in each category. Refer to school administrator (97.0%), confiscate substance (96.3%), and inform parents (88.8%) were the top three most frequently indicated first-offence violation responses and are all in the “other” category. Issue a warning was indicated the least frequently of the “other” approaches (36.6%). Out-of-school suspension (88.1%) and alert police (76.9%) were the next two approaches indicated by the most schools, both in the punitive consequence category. The other punitive approach, issue a fine, was the second least frequently indicated option overall (9.0%). The two mild approaches (assign help around the school and additional class work) were the other options indicated by the fewest schools (5.2% and 12.7%, respectively). Out of the three possible supportive approaches, *encouraging* participation in an assistance, education, or cessation program was selected by the most schools (69.4%), compared to 29.1% of schools that indicated *requiring* participation in a program. Almost half (48.5%) of schools indicated referring students to a counsellor as an option. For the two moderate approaches, in-school suspension was selected by more schools than detention (40.3% versus 26.1%).

Table 1 indicates the frequency of first-offence disciplinary approaches that were selected by secondary schools per province. The mean number of first-offence response options selected by schools was similar across provinces, from the highest mean of 7.44 (SD = 2.97, Range = 4–13) in BC to the lowest mean of 6.81 (SD = 2.03, Range = 3–14) in Ontario.

For punitive approaches, at least half of participating schools in Quebec (88.5%), Ontario (75.9%), BC (56.3%) and Alberta (50.0%), indicated alerting the police as a first-offence response option. Out-of-school suspension was selected by most schools in all provinces (84.6% of schools in Quebec to 91.4% in Ontario). Issue a fine was rarely selected across provinces, with no BC schools and only one participating Alberta school selecting this option.

For moderate approaches, a higher proportion of schools in Alberta indicated both in-school suspension (75.0%) and detention (62.5%), relative to schools in other provinces; however, the sample size in Alberta was limited to eight schools. Few Ontario schools indicated either moderate approach and only 2 of the 16 participating schools in BC indicated using detention.

In the supportive category, schools in BC selected encouraging participation in an assistance, education, or cessation program (81.3%) and referring the student to a counsellor (75.0%) more often in comparison to schools in other provinces. Encouraging participation in an assistance, education, or cessation program was selected by at least half of the schools in each province, while requiring participation in program was selected by relatively few schools in all provinces. Quebec schools indicated requiring participation in a program most frequently (36.5%) but refer to a counsellor (25.0%) least frequently, compared to participating schools in other provinces.

The two mild approaches were indicated by few schools across the provinces. No participating schools in Alberta selected assigning help around the school, while three BC schools (18.8%), and two schools in Quebec (3.8%) and in Ontario (3.4%) indicated this option. Assigning additional class work was selected more often by schools in Quebec (21.2%), compared to three schools in BC (18.8%), two in Ontario (3.4%), and one Alberta school (12.5%).

Within the other category, refer to an administrator, confiscate substance, and inform parents were indicated by most schools in all provinces. All participating schools in BC and Alberta indicated referring students to a school administrator, and confiscation of the substance was selected by all participating Alberta and Quebec schools. Quebec schools indicated issue a warning as a response option the most commonly (46.2%) and Alberta schools the least (12.5%). Informing parents was selected more often by schools in BC (93.8%) and least often in Ontario schools (87.9%).

Always using a progressive discipline approach, in which sanctions get stronger for subsequent violations, was reported by 89.7% of participating COMPASS secondary schools. All participating schools in Alberta indicated “always” using the progressive discipline approach, followed by schools in Quebec (90.4%), Ontario (90.0%), and BC (81.3%). The remaining schools indicating “sometimes” using the progressive disciplinary approach, as no schools indicated “never”.

Overall, based on the disciplinary approach style categories described, 46 schools were classified as Authoritarian, 49 as Authoritative, 18 as Neglectful, 15 as Permissive/Supportive, and 3 as Other. Table 2 displays the characteristics of schools categorized into the disciplinary approach styles. Schools lacking all school-level variables were excluded from this analysis, resulting in 131 schools included in the final classification approach.

In Ontario, almost half of the schools were classified as using an Authoritative disciplinary approach style (49.1%), followed by Authoritarian (26.3%); while in Quebec, 51.9% of schools were classified as Authoritarian. Quebec was the only province with schools in the Other category (5.8%). The most common disciplinary approach style in participating BC schools was Authoritative (37.5%), followed by Permissive/Supportive (28.6%). The eight Alberta schools were evenly divided into Authoritarian, Authoritative, Neglectful, and Permissive/Supportive.

School enrolment and urbanicity were similar across all categories of school disciplinary approach styles. The three schools that were classified as Other were in the lowest area median household income category. Schools with the highest area median household income of CDN 75,000+ were most often classified as Authoritative (52.9%). Among the 90.0% of schools that reported always using the progressive discipline approach, 37.5% were classified as Authoritative and 35.0% as Authoritarian.

## 4. Discussion

This study examined the current disciplinary environment associated with violating school substance use policies in the school year immediately post cannabis legalization in Canada. Most schools were classified as using Authoritarian or Authoritative disciplinary approach styles to student cannabis policy violations. A range of responses to first-offence violations was indicated by participating schools, with referring the student to a school administrator and confiscation of the substance the most frequently selected approaches, followed by informing parents and out-of-school suspension.

The majority of participating schools indicated they “always” use the progressive discipline approach, while the remainder indicated “sometimes”. Government policies for the required use of progressive discipline vary by province and region. Participating Ontario schools were most likely to indicate always using the progressive discipline approach, followed by Quebec. In Ontario, the Ministry of Education has mandated progressive discipline [21]; however, it should be noted that the Ontario government mandate varies from the measure used in this study, which only defines progressive discipline as employing stronger sanctions for subsequent violations of school substance use policy violations. The Ontario Ministry of Education policy extends further, advising that schools consider a range of consequences and supports, taking into account mitigating factors, such as the student’s history of misconduct, to determine the most appropriate response to each situation [20]. The Ontario Progressive Disciplinary Policy also advises schools to help students learn from their choices and to engage parents in an ongoing dialogue of students’ behaviour to ensure early and ongoing intervention [20]. In this study, 88% of Ontario schools reported engaging the students’ parents in regard to cannabis policy first-offence violations, down from 92% in the 2017/18 school year [22]. Further research is needed to identify the specific consequences used by schools that follow a progressive discipline approach, and how these consequences are decided upon to elucidate the most effective comprehensive strategies.

Based on the previous literature [7,17,22,28], the disciplinary response options for student first-offence violations of school cannabis policies were categorized as punitive, supportive, mild, moderate, and other approaches. Three “other” approaches were indicated most commonly by schools: referring the student to a school administrator, confiscation of the substance, and informing parents. It seems likely that referring students to the school administrator and confiscating the substance are first steps upon violating the school cannabis policies, in order to decide on a disciplinary approach. About one-third of schools indicated giving a warning as a first-violation response, which may reflect what occurs when no disciplinary approach is decided on after referral to an administrator. Out-of-school suspension was selected more frequently by participating schools than the previous year of the COMPASS study, immediately before cannabis legalization [22]. Previous research found the use of out-of-school suspensions for substance use policy violations to be associated with increased school-wide student substance use [29]. Suspensions have also been linked to disengagement from school, delinquency, and antisocial behaviours [18,19].

Alerting the police, another punitive approach, was the next most common discipline response, indicated by three-quarters of participating schools. More schools in Quebec indicated alerting the police from the options provided (89%), compared to about half to three-quarters of participating schools in other provinces. Alerting the police is not a federal legal requirement of schools, despite cannabis use remaining illegal among youth and on school properties. At the provincial level, Ontario principals are advised to report trafficking, but to consider mitigating factors when deciding whether to alert the police when a student is under the influence of a substance [21]. As there has been a movement away from stricter and zero-tolerance policies, it is possible that police are infrequently involved and remain among the set of options selected by schools from times when more punitive tactics were common. In support, fewer schools indicated alerting the police than in previous school year [22]. Future study waves will be needed to determine if this reduction indicates a movement away from police involvement by schools for responding to student substance use.

Overall, supportive first-offence disciplinary approaches were indicated less often than punitive options. Schools in BC indicated encouraging participation in an assistance, education, or cessation program more often than schools in other provinces, consistent with the previous school year [22]. Encouraging participation in a program was the more commonly used approach of the supportive options, both in the year preceding legalization [15] and the year immediately post-legalization; while requiring participation in a program was the least frequently indicated. Conflicting evidence exists on the effectiveness of mandated versus non-mandated substance use programs for adolescents [30,31,32]. Previous research suggests the use of supportive approaches promotes more positive outcomes for students; for example, referring students to a counsellor has been shown to decrease student substance use by almost 50% [8]. In Ontario, schools are encouraged to offer students supportive approaches as disciplinary actions and to build partnership with community-based service providers [20]. Few schools indicated using the mild first-offence disciplinary approaches of assigning students additional classwork and to help around the school. Limited literature has explored the effectiveness of these measures. Differences in punitive and supportive approaches across provinces potentially reflect the availability of resources for preventative programs and interventions, variations in the political climate, and/or social norms regarding cannabis. Historically, BC and Alberta tend to have more tolerant views of cannabis, as signified by the greater support for legalization [33], in comparison to Quebec, where the legal age was recently raised to 21 years old [34].

Based on previous literature on school disciplinary styles and Baumrind’s *Theory of Parenting* [13,18,30,35], a classification scoring approach for this study was developed. Based on the response options selected for student first-offence cannabis policy violations, schools were classified into five main disciplinary approach style categories: Authoritarian (high punitive, low support), Authoritative (high punitive, high support), Neglectful (low punitive, low support), Permissive/Supportive (low punitive, high support), and Other (a mixed approach). Several researchers have applied the theory of parenting styles to school climate and categorized schools based on student-reported measures of disciplinary demandingness and responsiveness [13,30,35]; however, no previous studies have used school-level measures to classify disciplinary environments.

Most schools were categorized as using either an authoritative or authoritarian approach to discipline related to student cannabis policy violations. Of the participating schools, 37% indicated at least two first-offence approaches corresponding to the high structure and responsiveness characterizing authoritative styles, while 36% of schools were categorized as using an authoritarian approach to discipline, which is defined in the literature as a strict, obedient, and “zero-tolerance” environment. The hypothesized benefit of using an authoritative approach is that students become more engaged when they are in a structured environment where they feel encouraged and supported [36]. Authoritative practices have been associated with lower truancy, delinquency, and dropout rates than authoritarian approaches [30,37]; however, in these studies, school disciplinary styles were classified according to student perceptions of responsiveness and demandingness, and not school protocols.

About one-tenth of participating schools were classified as permissive/supportive. Schools with a permissive/supportive disciplinary approach have been described as placing low demands on students [30]. When classified by student perceptions, this category has been linked to lower achievement and school engagement, and higher rates of problem behaviours [30]. Lastly, 14% of schools were classified into the neglectful category. Little research has been conducted on this school disciplinary approach category [15]. Baumrind’s Theory of Parenting describes neglectful styles as minimal effort, inconsistent, and often sporadic disciplinary practices [30].

### Strengths and Limitations

A key strength of the study is the school-level data from four Canadian provinces, as existing literature on school disciplinary styles is based on student report, without assessment of the approaches used by schools. Several limitations of this study warrant consideration. It is plausible that the school respondents were unaware of how policies are being implemented, although the survey is designed to be filled out by a staff member most knowledgeable about the programs and policies implemented within the school and respondents are encouraged to complete it as a group to gather the most accurate information. Additionally, study staff follow up with the schools regarding any unclear or missing responses by phone or email. It is important to note that the question assessing first-offence approaches used a “check all that apply” design, and therefore, the responses may not accurately reflect the usual consequences used by school administrators. That is, the data do not allow for analysis of how frequently each first-offence disciplinary approach is utilized by a school or under what circumstances (e.g., cannabis possession, use, or distribution) schools will choose to use specific consequences. It is likely that school administrators take mitigating factors into account when deciding on appropriate responses to cannabis policy violations. Future qualitative research should explore how these decisions are being made. Lastly, as COMPASS was not designed to be representative, a convenience sample of schools and boards was used based on permitted use of active information passive parental consent protocols for the collection for student-level data, which reduce school burden and collect more robust student substance use data [26,38].

## 5. Conclusions

Based on disciplinary options reported by schools for first-offence cannabis policy violations, most schools were classified as using authoritative and authoritarian disciplinary approaches, with fewer schools using neglectful, permissive/supportive, and other styles. All schools reported at least sometimes using progressive discipline, with the majority reporting always using this approach, yet most schools indicated punitive consequences as first-offence response options. Punitive first-offence consequences and the authoritarian disciplinary style do not align with progressive discipline, described as increasingly stronger sanctions with subsequent offences. It may be beneficial to offer schools and boards further support in implementing progressive discipline policies. Additionally, research on how school disciplinary styles relate to student substance use is an important next step. Future research will explore these categories in association with student perceptions of school supportiveness for the cessation/prevention of substance use, to offer insight into school contexts that may deter youth early onset or problematic cannabis use.

## Figures and Tables

**Figure 1 ijerph-18-02472-f001:**
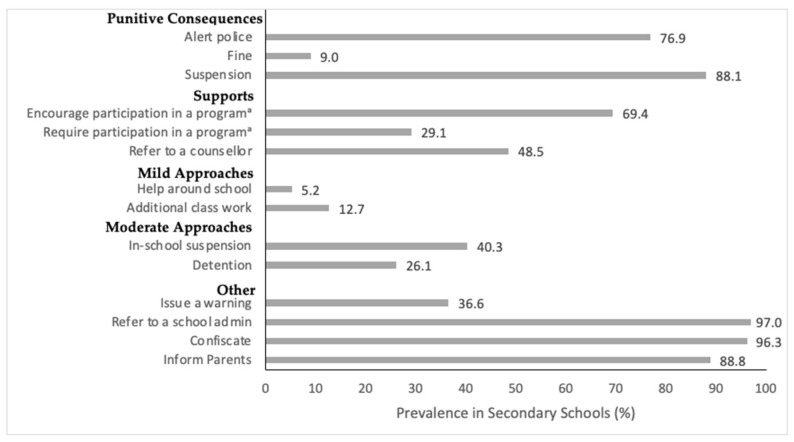
Percentage of Canadian secondary schools (n = 134) that indicated using various disciplinary response options for student first-offence violations of school cannabis policies in Year 7 (2018–2019) of the COMPASS study.^a^ Assistance, education, or cessation program. Note: the question used a “check all that apply” design.

**Table 1 ijerph-18-02472-t001:** Prevalence of disciplinary approach options selected by Canadian secondary schools (n = 134) for use in response to student violations of school cannabis policies by province in Year 7 (2018/19) of the COMPASS Study.

	BC (%)	Alberta (%)	Ontario (%)	Quebec (%)	Overall (%)
	N = 16	N = 8	N = 58	N = 52	N = 134
***First-Offence Disciplinary Approach***					
**Punitive options:**					
Alert police	56.3	50.0	75.9	88.5	76.9
Issue a fine	0	12.5	12.1	7.7	9.0
Suspension	87.5	87.5	91.4	84.6	88.1
**Supportive options:**					
Encourage participation in an assistance, education, or cessation program	81.3	50.0	65.5	73.1	69.4
Require participation in an assistance, education, or cessation program	25.0	12.5	25.9	36.5	29.1
Refer to a counsellor	75.0	62.5	96.6	25.0	48.5
**Mild options:**					
Assign to help around school	18.8	0	3.4	3.8	5.2
Assign additional class work	18.8	12.5	3.4	21.2	12.7
**Moderate options:**					
In-school suspension	50.0	75.0	19.0	55.8	40.3
Detention	12.5	62.5	13.8	38.5	26.1
**Other options:**					
Issue a warning	43.8	12.5	29.3	46.2	36.6
Refer to a school administrator	100	100	96.6	96.2	97.0
Confiscate substance	81.3	100	96.6	100	96.3
Inform parents	93.8	87.5	87.9	88.5	88.8
***Progressive discipline: Always***	81.3	100	93.1	90.4	90.3
***Progressive discipline: Sometimes***	18.8	0	8.6	9.6	9.7

**Table 2 ijerph-18-02472-t002:** Canadian secondary schools (n = 131) in Year 7 (2018/19) of the COMPASS Study classified into School Disciplinary Approach Styles.

		Authoritarian N = 46		Authoritative N = 49		Neglectful N = 18		Permissive/Supportive N = 15		Other N = 3	
		N	%	N	%	N	%	N	%	N	%
**Province**	BC (N = 14)	2	14.3%	5	35.7%	3	21.4%	4	28.6%	0	0.0%
	Alberta (N = 8)	2	25.0%	2	25.0%	2	25.0%	2	25.0%	0	0.0%
	Ontario (N = 57)	15	26.3%	28	49.1%	10	17.5%	4	7.0%	0	0.0%
	Quebec (N = 52)	27	51.9%	14	26.9%	3	5.8%	5	9.5%	3	5.8%
**School Enrolment**	0–500	22	35.5%	20	32.3%	10	16.1%	7	11.3%	3	4.8%
	501–1000	19	33.3%	24	42.1%	7	12.3%	7	12.3%	0	0.0%
	1001–1500	5	41.7%	5	41.5%	1	8.3%	1	8.3%	0	0.0%
**Urbanicity**	Rural/Small Urban	23	37.1%	23	37.1%	8	12.9%	6	9.7%	2	3.2%
	Medium/Large Urban	23	33.3%	26	37.7%	10	14.5%	9	13.0%	1	1.4%
**Area median household income**	CDN 25,000–75,000	39	40.2%	31	32.0%	11	11.3%	13	13.4%	3	2.3%
	CDN 75,000+	7	20.6%	18	52.9%	7	20.6%	2	5.9%	0	0.0%
**Progressive Discipline**	Always	42	35.0%	45	37.5%	15	2.5%	15	12.5%	3	2.5%
	Sometimes	4	36.4%	4	36.4%	3	27.3%	0	0.0%	0	0.0%

## Data Availability

COMPASS data are available for researchers upon request through successful completion and approval of the online COMPASS data usage application (https://uwaterloo.ca/compass-system/information-researchers, accessed on 2 March 2021).

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
