# Peer review of "Disciplinary Approaches for Cannabis Use Policy Violations in Canadian Secondary Schools"

_ijerph, 2021, doi:10.3390/ijerph18052472_

Round 1
Reviewer 1 Report
The article discusses disciplinary approaches for marijuana policy in Canadian high schools. The manuscript expands upon important knowledge. It is written and articulated well. However, I feel that there are things missing that could strengthen the paper further:
- Line 12 – if COMPASS is an acronym, you should define it here.
- Lines 38 to 40, RE "alienate students" – I think that this statement could benefit from the work of researchers who have looked at other school policies to demonstrate that authoritarian or paternalistic policies not only alienate students, but they also impact students’ “mental health” and wellbeing, thus constructing a powerful way to show how harmful punitive policies can be. It may be worthwhile to cite the work of researchers such as: Syed, I.U. (2012) Forced Assimilation is an Unhealthy Policy Intervention: the Case of the Hijab Ban in France and Quebec, Canada. International Journal of Human Rights. 17(3): 428-440. doi: 10.1080/13642987.2012.724678. Also noteworthy may be: S.V. Wayland, ‘Religious Expression in Public Schools: Kirpans in Canada, Hijab in France’. Ethnic and Racial Studies 20, no 3 (1997): 545–561.
- Line 97: If COMPASS is an acronym, you should write the full form here.
- Line 98: please delete “hereby referred to as” because the parentheses denote the short hand for this acronym.
- Line 106: (SPP) should be written after “Program” and the words “Scan Tool” should have lower case “s” and “t” because they are not part of the acronym.
- Lines 228 to 233 and Lines 252 to 255: Do you suppose there is an explanation for regional differences across Canada? Does this pattern reflect political climates in those regions?
- Line 251, RE “in any province”. Perhaps state “across the provinces”. Had it been the case that no mild approaches were indicated, you could leave “in any province” but it seems awkward when there are two mild approaches.
- Line 267, “indciating" should be “indicating”.
- Table 2 – For Ontario, the 4 under permissive/supportive is shifted left and does not align in the column.
- Line 417 to 418 – you should consider strengths of the paper along with the said limitations that you currently have. Perhaps rename the section “Strengths and limitations” thereafter.
If you decide to incorporate these revisions, please upload a manuscript that contains tracked changes or other method to highlight revisions. Thank you for the opportunity to review this work.
Author Response
Reviewer #1:
The article discusses disciplinary approaches for marijuana policy in Canadian high schools. The manuscript expands upon important knowledge. It is written and articulated well. However, I feel that there are things missing that could strengthen the paper further:
- Line 12 – if COMPASS is an acronym, you should define it here.
Response: Thank you for your feedback. This change has been updated within the document on lines 11-12.
- Lines 38 to 40, RE "alienate students" – I think that this statement could benefit from the work of researchers who have looked at other school policies to demonstrate that authoritarian or paternalistic policies not only alienate students, but they also impact students’ “mental health” and wellbeing, thus constructing a powerful way to show how harmful punitive policies can be. It may be worthwhile to cite the work of researchers such as: Syed, I.U. (2012) Forced Assimilation is an Unhealthy Policy Intervention: the Case of the Hijab Ban in France and Quebec, Canada. International Journal of Human Rights. 17(3): 428-440. doi: 10.1080/13642987.2012.724678. Also noteworthy may be: S.V. Wayland, ‘Religious Expression in Public Schools: Kirpans in Canada, Hijab in France’. Ethnic and Racial Studies20, no 3 (1997): 545–561.
Response: Thank you for bringing this meaningful work to our attention. These papers have been included, and citated on lines 49-50, references 11 and 12.
- Line 97: If COMPASS is an acronym, you should write the full form here.
Response: Thank you, the full form has updated on line 100-101.
- Line 98: please delete “hereby referred to as” because the parentheses denote the short hand for this acronym.
Response: “hereby referred to as” has been removed from the document on line 101.
- Line 106: (SPP) should be written after “Program” and the words “Scan Tool” should have lower case “s” and “t” because they are not part of the acronym.
Response: Thank you, this has been updated on line 113 in the document.
- Lines 228 to 233 and Lines 252 to 255: Do you suppose there is an explanation for regional differences across Canada? Does this pattern reflect political climates in those regions?
Response: The reviewer raises an interesting potential explanation, which points to one of the reasons we were interested in exploring disciplinary styles across provinces. The differences potentially reflect the stricter political climate in terms of cannabis in Quebec and Ontario relative to British Columbia and Alberta.
To reflect this comment, we have added to the revised manuscript on lines 326-330: “Differences in punitive and supportive approaches across provinces potentially reflect the availability of resources for preventative programs and interventions, variations in the political climate, and/or social norms regarding cannabis. Historically, BC and Alberta tend to have more tolerant views of cannabis, as signified by the greater support for legalization (35), in comparison to Quebec, where the legal age was recently raised to 21 years old (36).”
- Line 251, RE “in any province”. Perhaps state “across the provinces”. Had it been the case that no mild approaches were indicated, you could leave “in any province” but it seems awkward when there are two mild approaches.
Response: Thank you, “across the provinces” has been updated in the document on line 237.
- Line 267, “indciating" should be “indicating”.
Response: Thank you for identifying this error.
- Table 2 – For Ontario, the 4 under permissive/supportive is shifted left and does not align in the column.
Response: Thank you, the alignment has been fixed in Table 2.
- Line 417 to 418 – you should consider strengths of the paper along with the said limitations that you currently have. Perhaps rename the section “Strengths and limitations” thereafter.
Response: Thank you for your comment. This section has been updated to “Strengths and Limitations”.
Reviewer 2 Report
Thank you for the opportunity to review this manuscript. This manuscript examines disciplinary approaches used in Canadian secondary schools for students who violate school cannabis policies. Overall, the study is straightforward, well described, and presents findings that would be of interest to the journal’s audience. The suggestions are offered in the spirit of strengthening the manuscript.
ABSTRACT:
While abstract generally follows the format of a structured abstract as instructed by the journal, I would suggest the conclusion be revised; the study being “the first to classify school discipline approach styles using school-level measures” does not indicate the main conclusion or interpretation that should be drawn from this study.
INTRODUCTION:
I would suggest that introduction would benefit from some additional context on trends/prevalence of youth cannabis use as well as the context of legalization in Canada (e.g. current legal age of purchase/possession, prohibition on school grounds, previous legislation in provinces?)
METHODS:
Design: The authors provide references for the full description of the study design but I think how schools are recruited is an important detail to include in the methods.
Measures: This section seems more appropriate in the introduction or discussion, “To the best of our knowledge, no study has classified school disciplinary approach styles based on the disciplinary polices or procedures assessed by school-level data. Therefore, this study is the first to establish a framework for classifying schools into disciplinary approach styles based on first-offence response measures reported by schools.”
RESULTS
The first sentence seems more appropriately placed in the methods.
In Table 1, its unclear what the footnote (a) next to the (%) in the Ontario column is for?
Table 2 offers some school characteristics (i.e., enrollment, urbanicity, median household income) by disciplinary approach; however, it would be helpful to provide these school characteristics for the entire sample to understand if or how much this sample may differ from secondary schools overall.
DISCUSSION
I think the writing of the discussion can be tightened a bit as many results are repeated here but overall, it interprets the results in the context of the existing literature and policy environment. Overall, the conclusions are reasonable and limitations are addressed.
Typo in line 455:
Abbreviations: COMPASS: Cannabis use, Obesity, Mental nealth, Physical activity, Alcohol use,
Author Response
Thank you for the opportunity to review this manuscript. This manuscript examines disciplinary approaches used in Canadian secondary schools for students who violate school cannabis policies. Overall, the study is straightforward, well described, and presents findings that would be of interest to the journal’s audience. The suggestions are offered in the spirit of strengthening the manuscript.
ABSTRACT:
While abstract generally follows the format of a structured abstract as instructed by the journal, I would suggest the conclusion be revised; the study being “the first to classify school discipline approach styles using school-level measures” does not indicate the main conclusion or interpretation that should be drawn from this study.
Response: Thank you for your feedback. The conclusion of the abstract has been revised to better reflect the conclusions and interpretations from the study.
INTRODUCTION:
I would suggest that introduction would benefit from some additional context on trends/prevalence of youth cannabis use as well as the context of legalization in Canada (e.g. current legal age of purchase/possession, prohibition on school grounds, previous legislation in provinces?)
Response: Thank you. On lines 35-44, the introduction has been updated to discuss additional context on youth use, and legalization in Canada: “Bill C-45 set a nationally regulated provision on cannabis production, distribution, retail, possession and consumption, however, provinces and territories were still tasked with developing additional legislations and policies (2). Even though cannabis use is only legal among adults (18 or 19, or 21years and over, depending on the province or territory), youth may be indirectly affected through de-stigmatization of use and potential changes in accessibility (3) (4). Youth in Canada use cannabis at the highest rates globally (5). After a steady decline in youth cannabis use, a gradual increase has since occurred from the beginning of federal discourse around legalization, particularly in intermittent use (6)”
METHODS:
Design: The authors provide references for the full description of the study design but I think how schools are recruited is an important detail to include in the methods.
Response: To further describe the recruitment processes, the statement “School boards were purposefully selected based on permitted use of active-information passive parental permission consent protocols for the collection of student-level data. Information letters and recruitment packages were sent to school boards, and then to individual schools, after board approval and according to the requirements outlined by their board (24)” has been added to lines 103-104. The technical report, “School Board and School Recruitment Procedures” citated provides further details.
Measures: This section seems more appropriate in the introduction or discussion, “To the best of our knowledge, no study has classified school disciplinary approach styles based on the disciplinary polices or procedures assessed by school-level data. Therefore, this study is the first to establish a framework for classifying schools into disciplinary approach styles based on first-offence response measures reported by schools.”
Response: Thank you for the feedback. This section has been moved to the introduction, on lines 96-99.
RESULTS
The first sentence seems more appropriately placed in the methods.
Response: This sentence has now been moved to the methods, on lines 151-152.
In Table 1, its unclear what the footnote (a) next to the (%) in the Ontario column is for?
Response: Thank you for correcting this missing footnote. The footnote has been removed from the table.
Table 2 offers some school characteristics (i.e., enrollment, urbanicity, median household income) by disciplinary approach; however, it would be helpful to provide these school characteristics for the entire sample to understand if or how much this sample may differ from secondary schools overall.
Response: Thank you for your feedback. At the beginning of the results section, on lines 179-181, the following description for the entire sample has been added: “Out of the 134 schools, the majority were classified as having an enrolment within 0-500 (47.8%). The most common school area median household income was $25,000-$75,000 (74.63%) and most schools fell within a medium/large urbanicity range (93.0%).”
DISCUSSION
I think the writing of the discussion can be tightened a bit as many results are repeated here but overall, it interprets the results in the context of the existing literature and policy environment. Overall, the conclusions are reasonable and limitations are addressed.
Response: Thank you for your feedback. The discussion has been revised accordingly to minimize repeated results.
Typo in line 455:
Abbreviations: COMPASS: Cannabis use, Obesity, Mental nealth, Physical activity, Alcohol use,
Response: Thank you for identifying this error.
Round 2
Reviewer 1 Report
The article discusses disciplinary approaches for marijuana policy in Canadian high schools. The manuscript expands upon important knowledge. It is written and articulated well. The authors have addressed all of the reviewer’s concerns. I feel that the paper should be accepted for publication. Congratulations, and thank you for the opportunity to review this work.